# Role of Oligodendrocytes and Myelin in the Pathophysiology of Autism Spectrum Disorder

**DOI:** 10.3390/brainsci10120951

**Published:** 2020-12-08

**Authors:** Alma Y. Galvez-Contreras, David Zarate-Lopez, Ana L. Torres-Chavez, Oscar Gonzalez-Perez

**Affiliations:** 1Department of Neuroscience, Centro Universitario de Ciencias de la Salud, University of Guadalajara, Guadalajara 44340, Mexico; 2Laboratory of Neuroscience, School of Psychology, University of Colima, Colima 28040, Mexico; dzarate@ucol.mx (D.Z.-L.); torres_chavez@ucol.mx (A.L.T.-C.); 3Physiological Sciences PhD Program, School of Medicine, University of Colima, Colima 28040, Mexico

**Keywords:** oligodendrogenesis, myelination, epidermal growth factor, insulin-like growth factor

## Abstract

Autism Spectrum Disorder (ASD) is an early neurodevelopmental disorder that involves deficits in interpersonal communication, social interaction, and repetitive behaviors. Although ASD pathophysiology is still uncertain, alterations in the abnormal development of the frontal lobe, limbic areas, and putamen generate an imbalance between inhibition and excitation of neuronal activity. Interestingly, recent findings suggest that a disruption in neuronal connectivity is associated with neural alterations in white matter production and myelination in diverse brain regions of patients with ASD. This review is aimed to summarize the most recent evidence that supports the notion that abnormalities in the oligodendrocyte generation and axonal myelination in specific brain regions are involved in the pathophysiology of ASD. Fundamental molecular mediators of these pathological processes are also examined. Determining the role of alterations in oligodendrogenesis and myelination is a fundamental step to understand the pathophysiology of ASD and identify possible therapeutic targets.

## 1. Introduction

In 1943, Kanner described the first symptoms of Autism Spectrum Disorder (ASD) as an innate disturbance of affective contact. In 1944, Hans Asperger included a psychopathic disturbance of social interaction in the symptomatology of ASD [1]. To date, ASD is considered an early neurodevelopmental disorder [1], which is characterized by substantial deficits in social interaction and communication associated with repetitive and restricted behaviors [2]. The incidence of ASD around the world is 1 per 160 children [3] but some epidemiological variations have been reported among world regions [3,4]. Possible explanations for this variability include dissimilarity in methods, variations in diagnostic or community identification, and potential risk factors [5]. To date, the Diagnostic and Statistical Manual of Mental Disorders (DSM-5), ASD comprises only two symptomatic domains: deficits in communication and social interaction, and repetitive and restricted behavior [6]. Thus, the current ASD definition includes autistic disorder, Asperger’s disorder, pervasive developmental disorder not otherwise specified (PDD-NOS), Rett’s disorder, and childhood disintegrative disorder, which represent a moderate variation in ASD diagnosis with respect to the previous DSM version [7]. Throughout life, the clinical course of ASD is variable and includes a wide range of clinical manifestations [8], including poor social skills, language, intellectual disabilities, sensory abnormalities (hyper- or hypo-sensory responsiveness), motor tics, and gross motor discoordination [8,9,10,11]. Nevertheless, ASD is a heterogeneous disorder with many inter-subject dissimilarities in social behaviors that some authors have associated with a combination of genetic variants that contribute to different phenotypic outcomes [12,13].

During the first years of life, the clinical symptoms correlate with two pathophysiological abnormalities in the brain. The overgrowth of certain brain regions precedes some symptoms of ASD and also coincides with periods of high neural plasticity [14,15]. From prenatal stages to birth, a slight delay in neural development is observed, which seems to be compensated by an accelerated brain growth during the first years of life [16,17]. This rapid overgrowth tends to reduce during the second and third years of life [14]. From the 5th to 16th year of life, this abnormal brain growth ceases and tends to normalize when compared to typically developing subjects [14,18]. In adults, some structures, such as the frontal lobe, cerebellum, and amygdala, show a normal or reduced volume as compared to typically developing subjects [19,20,21,22]. Recently, it has been reported that children with ASD show significant alterations in the connectivity between the right insula with the supramarginal gyrus and left superior frontal gyrus [23]. Adolescents with ASD also show age-dependent alterations in the connectivity between the frontal lobe and parietal region [24], which supports the notion that ASD is a dynamic developmental syndrome. Some of these anatomical abnormalities have been associated with extensive changes in myelination, excessive oxidative stress [25], glial activation [26], minicolumn pathology [27], abnormal neurogenesis, and neuronal migration [28,29]. Because the cerebral size in patients with ASD often correlates with functional deficits [30,31], these changes in the brain-growth pattern seem to be crucial to understand the symptomatology and etiology of ASD.

In the postnatal brain, white matter enlargement is one of the biological events that produces considerable changes in the volume of brain regions [32]. In the human brain, myelination is mostly a postnatal process that reaches its highest during childhood and continues until early adulthood [33]. Hence, myelination and oligodendrogenesis appear to be crucial events that notably modify several neuroanatomical structures. Increasing evidence suggests that this dynamic process may be disrupted in the ASD brain [34]. Patients with ASD show enlarged cerebellar and cerebral white matter structures and this increase occurs during the first years of life. This brain overgrowth of white matter observed in ASD is not sustained throughout life and tends to be smaller with age [14]. A pilot study in children diagnosed with autism between 4 and 6 years suggested that alterations in axons and myelin of the *corona radiata* appear to be associated with the clinical severity of ASD [35]. Additional evidence in adolescents indicates that ASD brains show fewer axons, less axonal volume, and a low density of white matter tracts in the corpus callosum, frontal-occipital fasciculus, right uncinate fasciculus, and right arcuate fasciculus [36]. Interestingly, pathological changes in the cytoarchitecture of white matter in all brain lobes [30] and altered myelination rate in corpus callosum [1,37] may explain the dysfunctional connectivity found in medial parietal and temporoparietal regions [16]. These structural disruptions have also been correlated to parental age [38] and may explain the severity of stereotypes and deficits in social interaction [31]. In contrast, clinical data indicate that the improvement in ASD symptoms is associated with white matter recovery [39]. In this review, we described the general pathogenesis and neuroanatomical changes found in several regions of the ASD brain and many of these areas correspond to white matter regions. Thus, we explained proteins, genes, and signaling pathways that regulate the oligodendroglial process, which in some cases are coincidentally affected in ASD. Finally, we compiled the most recent evidence that suggests a role for myelination and oligodendrocytes in the pathogenesis of ASD. Taken together, these reports suggest that some pathological changes observed in the white matter may explain the abnormal brain development related to ASD.

## 2. Pathophysiological Basis of ASD

ASD is considered by some authors as a brain connectivity disorder that, in turn, modifies the inhibition/excitation balance. Some reports sustain that the neuronal hypoconnectivity in frontal areas and fusiform face area observed in ASD patients [40] may explain the morphological alteration in brain size and interhemispheric connections [41,42,43]. This balance between neuronal inhibition and excitation is coordinated by several biological and molecular processes that modify synapse structure and brain plasticity [44], which may affect the frontotemporal, frontolimbic, frontoparietal, and interhemispheric connections [21]. Intriguingly, some local neuronal circuits in the frontal lobe are overconnected, whereas long-range connections (cortico-parietal, sub-cortical systems, and inter-hemispheric tracts) seem to be reduced in ASD patients [16,21,45,46].

However, abnormalities in neuronal connections per se cannot explain all the functional changes observed in ASD, and emerging evidence suggests that glial cells may play a pivotal role in neuroanatomical and behavioral changes found in ASD. Astrocytes and oligodendrocytes may contribute to neurochemical imbalances described in the autistic brain by disrupting neurotransmission or modifying axonal conduction. In mice models for ASD, the genetic depletion of glutamate transporter-1 (GLT-1) in astrocytes increases excitatory neurotransmission that is related to a high frequency of repetitive behaviors [44], whereas phosphatase and tensin homolog (PTEN) mislocation produces precocious maturation of oligodendrocytes that is associated with aberrant myelination [47]. A recent report suggests that a mutation in the eukaryotic translation initiation factor 4E (eIF4E) in microglia produces sex-dependent ASD-like behaviors by modifying synaptic development and function in male mice [48]. Therefore, neuroinflammation and pro-inflammatory cytokines released by microglia can alter gliotransmission, ion-channel expression, brain plasticity, and oxidative stress that may also lead to behavioral dysfunctions [49]. Interestingly, oxidative stress has also been implicated in the pathophysiological process of ASD by affecting the myelination process [50,51].

In this regard, oxidant radicals can damage the oligodendrocyte population that fails to differentiate into myelin-forming mature oligodendrocytes and increase a significant proliferation of oligodendrocyte precursor cells (OPCs) that, in turn, impair the whole myelination process [52]. Thus, oligodendrocytes are a very susceptible cell lineage to oxidant radicals because they have low levels of glutathione (a highly efficient antioxidant molecule) and elevated amounts of sphingolipids [53,54]. Therefore, oligodendrocytes represent a plausible cellular target for the oxidative stress identified in the ASD brain.

Abnormal development in the white matter and myelination has been found in animal models of ASD [55,56,57,58,59] and humans [60,61]. Neuroimaging studies support the notion that ASD patients have a complex and dynamic disorder [62] in which the level of white matter involvement is associated with clinical severity [60,63,64,65,66,67,68,69]. ASD patients show extensive alterations in the white matter of several cortical and subcortical regions, such as the orbitofrontal cortex [70], anterior cingulate cortex and lateral prefrontal cortex [61], external capsule [71,72], arcuate fasciculus [73,74,75], ventral temporal gyrus [60,76], temporo-parietal junctions [60,72,77], amygdala [60], occipitofrontal fasciculus [60], occipitotemporal gyrus [77], thalamus [78], superior and middle cerebellar peduncle [79,80], medial lemniscus [81], and the corticospinal tract [82,83,84] (Figure 1). However, some clinical reports have shown contradictory findings in the myelination process of ASD brains. From childhood to adolescence, the inferior longitudinal fasciculus shows pathological changes in the white matter density [83,84,85,86,87,88,89,90] (Table 1). The uncinate fasciculus of ASD patients younger than 15 years may show increased or decreased white matter density [64,65,74,75,91], but, after age 15, only an increase in the white matter has been found [87]. ASD patients older than 15 years may also have high white matter density in the lingual gyrus [78], amygdala-fusiform pathway, and left hippocampi-fusiform pathway [92].

Although a reduction in the white matter density and impaired structural integrity of corpus callosum appears to be a consistent finding throughout the clinical evolution of ASD [64,71,72,74,77,79,82,83,84,99,100,101,102], some clinical studies have found an increase in the genu and midbody subregion of the corpus callosum [94] or the entire corpus callosum [78]. This evidence indicates that white matter alterations might not be present in the whole white matter or restricted to certain portions of the corpus callosum [89,110,111]. The disparity in these clinical findings may be due to multiple factors, including age range, specific diagnosis, comorbidities, medication, gender, type of imaging, or sample size. Other factors such as inter-individual differences, nurture, and epigenetic factors can also explain the discrepancy among clinical studies. Nevertheless, these differences also suggest that ASD might be associated with abnormalities in the myelination process, which is a very dynamic event. Therefore, multicenter longitudinal studies are required to fully establish which brain regions are more susceptible to determine whether alterations in certain brain regions may be linked to some specific ASD phenotypes.

## 3. Role of Oligodendrocytes in ASD: Cellular and Molecular Evidence

In animal models of ASD, reduction in the proliferation of oligodendroglial cells and low levels of myelin basic protein (MBP) has also been implicated in ASD pathogenesis [55]. Oligodendrocytes are glial cells that myelinate the brain and spinal cord to insulate axons electrically and provide neurons with trophic and metabolic factors. Mature oligodendrocytes originate from OPCs that, throughout life, preserve the population of myelinating oligodendrocytes [112]. OPCs are a persistent cell population that consists of around 5% to 10% of the total number of cells in the adult brain [113,114]. White-matter regions contain a high number of oligodendrocytes and OPCs that are in contact with axons, which facilitate neuronal communication [115]. Migrating and resident OPCs express the plated-derived growth factor receptor alpha (PDGFRα) and the oligodendrocyte transcription factor 2 (Olig2), and NG2 proteoglycan (Figure 2), which are cell markers of these progenitor cells [29,116]. Remarkably, a subpopulation of NG2-expressing cells (NG2 glia) establish synaptic junctions with local neurons and modulates axonal conduction by releasing glutamate and stimulating α-Amino-3-hydroxy-5-methyl-4-isoxazole propionic acid (AMPA) receptors [117,118,119]. Experimental evidence has suggested that disruption in neuron–glia interactions promotes autistic-like features [120]. However, it is unknown whether NG2 glia or glial transmission is implicated directly in the pathophysiology or clinical manifestations of ASD patients.

The aberrant development of white matter found in patients with ASD (Figure 1) and disrupted protein levels in oligodendrocytes in ASD mice models (Figure 2) are suggestive of disruptions in the maturation of oligodendrocytes [40], which may convey changes in the white matter of corpus callosum and other white matter regions [21,36,40]. Deletions and duplications in the chromosome 15q11.2 region have been described in autistic patients and associated with alterations in myelination and abnormal development of the corpus callosum [125]. In humans, deletion carriers of 15q11.2 present a reduced volume of white matter [126]. This chromosomal region encodes CYFIP1 (Cytoplasmatic FMRP interacting protein 1), a protein that regulates cytoskeletal dynamics and protein translation. In mice, *Cyfip1* mutation reduces the myelin thickness and impairs neuronal connectivity in the corpus callosum that, in turn, produces aberrant behaviors and poor motor coordination in these mice [127]. Interestingly, deletions in chromosome 7 (7q11.23) have been associated with Williams Syndrome, a neurodevelopmental genetic disorder characterized by hypersociability and higher empathy [128,129], which has been also related to a decrease in the volume of white matter [130]. Therefore, disruptions in the myelination process could be associated with impairments in different chromosomes that, in turn, produce deficient electrical transmission resulting in impaired neurotransmitter release and behavioral manifestations [115].

Chromodomain helicase DNA binding proteins 7 (CHD-7) and 8 (CHD-8) are nucleosome remodeling factors that spatially and temporally control gene expression. CHD-7 is highly expressed in myelinating oligodendrocytes and its deficit or malfunction strongly compromises the myelination and remyelination process [131]. Transgenic *Chd7* mice demonstrate that this nucleosome remodeling factor is necessary throughout life for the transcription of *Sox10*, *Nkx2-2*, and *Gpr17* genes that regulate oligodendrocyte differentiation, and its inactivation decreases OPC survival and differentiation in cortex and corpus callosum via cellular tumor antigen p53 (Trp53) [131]. In contrast, CHD-8 is highly expressed in the early prenatal period and progressively decreases at later stages of development [132], but it remains highly expressed in the OPCs of adult white matter in the corpus callosum, optic nerve, and spinal cord [133]. CHD-8 regulates several genes associated with neurodevelopmental and synaptic functions that are affected in autism [134]. Interestingly, a clinical report found that *CHD8* mutations seem to be responsible for some phenotypic characteristics: macrocephaly and wide occipitofrontal circumference (head with a relatively large occipitofrontal diameter) and clinical behaviors (anxiety and social deficit) that have been associated with ASD [132,134]. A study in human neural stem cells and mid fetal human brain shows that the *CHD8* remodeling factor together with *ANK2*, *CUL3*, *DYRK1A*, *GRIN2B*, *KATNAL2*, *POGZ*, *SCN2A*, and *TBR1* are considered genetic risk factors to develop ASD [135]. Hence, this evidence suggests that *CHD7*/*CHD8* genes not only regulate the oligodendroglia differentiation and myelination but also provide some phenotypical features observed in people with ASD (macrocephaly, sleep dysfunction, growth retardation, and intellectual disability) [133,136].

Several growth factors regulate the proliferation and maturation of OPCs in the developing and adult brain. At the same time, systemic and local alterations in growth factors appear to be involved in the progression and severity of some psychiatric disorders [137]. Low levels of insulin-like growth factor 1 (IGF-1) reduce oligodendroglia survival [138] and cause OPCs to fail to differentiate into mature oligodendrocytes via bone morphogenetic protein (BMP) activation by up-regulating Noggin, Smad6, and Smad7 [139]. *Igf1* knockout mice show a reduction in the volume of the corpus callosum and anterior commissure that is associated with defectively myelinated axons and a decrease in the oligodendrocyte population [140]. Postnatally, these subjects also show fewer OPCs and mature oligodendrocytes that correlate with reduced expression of myelin proteins (MBP and the myelin proteolipid protein-PLP) [141]. Interestingly, children with ASD show low levels of IGF-1 in cerebrospinal fluid as compared with typically developing subjects [121,122]. In addition, low levels of IGF-1 are associated with deficient myelination and disorganization of neuronal circuits [142]. These pathological changes at the early stages of neurodevelopment may explain the impairment in synaptic development, inappropriate nerve conduction, and deficient axonal myelination frequently observed in ASD [142]. Therefore, IGF-1 and its signaling pathway are crucial for adequate axonal myelination and oligodendrocyte differentiation, but, when disrupted, white matter alterations, learning deficits, and ASD-like behaviors arise. To date, there are some experimental approaches to target the IGF-1 deficiency associated with ASD. The administration of IGF-1 for two weeks in Shank3-deficient mice, an ASD mouse model, notably improves the neuronal function by enhancing the long-term potentiation (LTP) and decreasing stereotypical behaviors [143]. Therefore, IGF-1 can be considered a promising therapeutic target, but further research is needed before using it in patients with ASD.

The Epidermal Growth Factor (EGF) and its receptors (ErbB 1–4) help preserve the oligodendrocyte population and repair demyelinating lesions by promoting proliferation, migration, and differentiation of OPCs in the adult brain [144,145,146,147]. Low levels of EGF and HER1 (ErbB1 in rodents) have been found in adults with high-functioning ASD [123]. In children, EGF levels correlate negatively with hyperactivity, tiptoe walking, and other motor signs [124]. Furthermore, the high expression of HER1 in children is correlated with high symptom severity (hyperactivity, conversational language, attention, eye contact, sound sensitivity, and expressive language) [148]. The interaction between neuregulin and ErbB4 protein, another member of the EGFR family, modifies the excitability of GABAergic neurons by increasing the inhibitory transmission [44,149]. GABAergic interneurons regulate cortical plasticity and cognitive flexibility in the frontal or parietal cortex [44]. Although the role of EGF in ASD has not been directly determined, this evidence suggests that changes in the ErbB family members or their ligands can modify the cortical plasticity and myelination as observed in ASD.

Both EGF and IGF-1 activate the PI3K/Akt/mTOR signaling pathway that regulates several intracellular functions, including cell growth, proliferation, differentiation, motility, survival, metabolism, and protein synthesis [150]. Akt/mTOR hyperactivity is commonly found in T cells [151] and peripheral blood [152] of patients with ASD. This hyperphosphorylation can, in turn, generate a deficiency of PTEN protein, a key negative regulator of the PI3K/Akt pathway [151], which results in Akt overactivation and increases the activity of mTOR [151,153]. The long-term activation of the Akt pathway prevents glutamate-mediated apoptosis in immature oligodendrocytes, which may induce aberrant myelination patterns [154]. Remarkably, patients with ASD and macrocephaly commonly show *PTEN* mutations [155]. In mice, disruption of PTEN activity by increasing Akt phosphorylation at Ser437 produces severe changes in the population of OPCs and in the genes and proteins involved in myelination (MBP, PLP, and myelin-associated glycoprotein-MAG), which induces social deficits that mimic some symptoms of ASD [156], such as increased anxiety and reduced social interest [153]. Interestingly, these animals also show aberrant myelin deposits adjacent to axons, increased volume of the corpus callosum, and brain enlargement [47,156,157]. Suppression of Akt/mTOR signaling improves ASD-associated symptoms in *Pten* knockout mice [153]. Therefore, *PTEN* alterations may be implicated in the pathogenesis of white matter abnormalities and behavioral symptoms observed in ASD [47,156,157]. A possible explanation for the low levels of IGF-1 and EGF reported in patients with ASD, despite the hyperactivity of the AKT/mTOR pathway, could be the interaction of receptors with other ligands as a compensatory mechanism, for instance, the IGF receptor (IGF1R) also interacts with insulin and insulin-like growth factor 2 (IGF-II) [158], whereas the EGFR has an affinity for other ligands such as TGF-ɑ and HB-EGF [159].

MAPK/ERK is another signaling pathway activated by IGF-1 and EGF [160] that regulates the differentiation, migration, proliferation, and survival of oligodendrocytes in the adult brain [161]. The MAPK/ERK signaling pathway also determines the morphology of oligodendrocytes [161] and regulates myelin synthesis [162] by promoting the activation of transcriptional factors (ELK1, AP2-complex, and CREB) that are necessary for the expression of MBP [162]. Interestingly, the crosstalk between PI3K/Akt/mTOR and MAPK pathways [163] controls the synthesis of oligodendroglia proteins [164] and OPCs proliferation [165]. However, while the role of MAPK/ERK in myelination and oligodendrocytes is well known, there is not enough evidence to clearly define the link between this signaling pathway and ASD. Hence, further research is required to elucidate this question.

## 4. Conclusions

ASD is a clinically heterogeneous disorder that seems to have a multifactorial origin. Typically, ASD has been considered as a brain connectivity disorder that produces a disparity in the inhibition/excitation balance. However, emerging evidence suggests that oligodendroglial cells play a role in the etiology of ASD. Oligodendrocytes insulate axons electrically and provide neurons with trophic factors that guarantee proper neurotransmission and axonal conduction in the white matter. Patients with ASD show an aberrant growth pattern in several white-matter regions that varies throughout life. During the first 15 years of life, axon tracts in the arcuate fasciculus, occipitofrontal fasciculus, and external capsule show a significant hypomyelination, whereas the fusiform gyrus and hippocampal area show a substantial increase in their myelin volume (Figure 1). In both cases, these alterations tend to normalize throughout development. Intriguingly, other brain regions such as cerebellum, cingulum, and internal capsule remain hypomyelinated. This atypical myelination pattern has been linked to several factors, including gene disruption and epigenetic factors. Some genes and growth factors implicated in this abnormal myelination process include Olig1, Olig2, *Sox10*, *PGFRA*, *Nkx2-2*, *Gpr17*, IGF-1, and EGF. Hence, this emerging evidence supports the notion that oligodendrogenesis and neural myelination may play a pivotal role in the pathophysiology of ASD and its clinical presentation. Understanding the molecular mediators involved in abnormal myelination patterns in the ASD brain may represent an initial step to design therapeutic targets that help improve social skills and disruptive behaviors observed in these patients.

## Figures and Tables

**Figure 1 brainsci-10-00951-f001:**
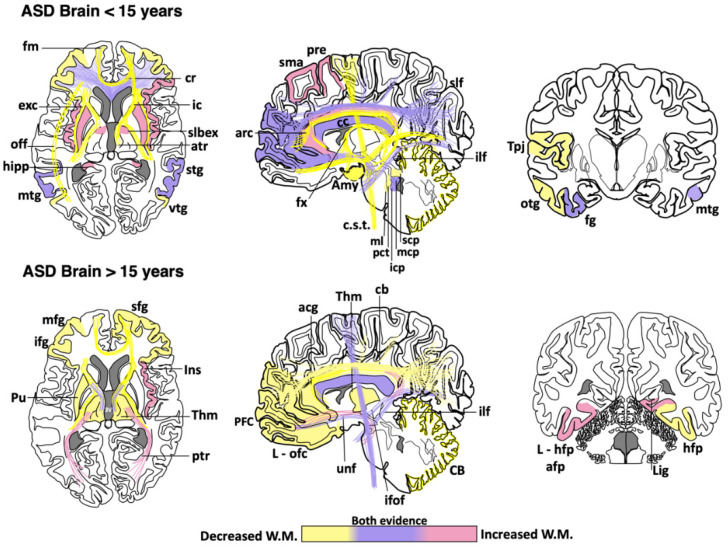
Cortical areas, subcortical areas and white matter tracts affected in patients with Autism Spectrum Disorder (ASD). White matter areas with the most noticeable changes are represented in the three anatomical planes of the human brain: axial, sagittal, and coronal, respectively. Brain schemes also indicate the most common myelination changes that occur in ASD patients during two different stages of development [21,30,36,60,64,65,67,70,71,72,73,74,75,76,77,78,79,80,81,82,83,84,85,86,87,88,89,90,91,92,93,94,95,96,97,98,99,100,101,102,103,104,105,106,107,108,109]. acg: anterior cingulate gyrus; afp: amygdala-fusiform pathway; Amy: amygdala; atr: anterior thalamic radiation; arc: arcuate fasciculus; CB: cerebellum; cb: cingulum bundle; cc: corpus callosum; c.s.t.: corticospinal tract; cr: corona radiata; exc: external capsule; fm: forceps minor; fx: fornix; fg: fusiform gyrus; hipp: hippocampus; icp: inferior cerebellar peduncle; ifof: inferior fronto-occipital fasciculus; ifg: inferior frontal gyrus; Ins: insula; ic: internal capsule; L-hfp: left hippocampus-fusiform pathway; L-ilf: Left inferior longitudinal fasciculus; L-ofc: left orbitofrontal cortex; lig: lingual gyrus; ml: medial lemniscus; mcp: middle cerebellar peduncle; mtg: middle frontal gyrus; mtg: middle temporal gyrus; off: occipitofrontal fasciculus; otg: occipitotemporal gyrus; pct: pontine crossing tracts; ptr: posterior thalamic radiation; pre: precentral area; PFC: prefrontal cortex; Pu: putamen; slbex: sub-lobar extranuclear area; scp: superior cerebellar peduncle; sfg: superior frontal gyrus; slf: superior longitudinal fasciculus; stg: superior temporal gyrus; sma: supplementary motor area; Tpj: temporoparietal junction; Thm: thalamus; unf: uncinate fasciculus; vtg: ventral temporal gyrus.

**Figure 2 brainsci-10-00951-f002:**
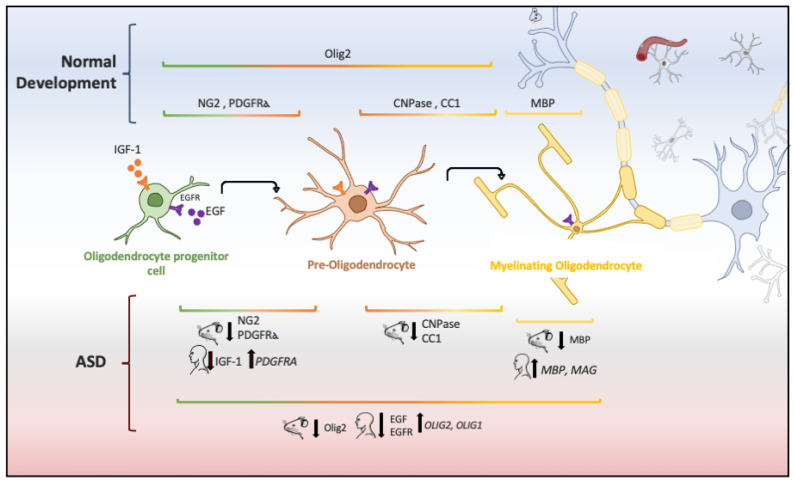
Oligodendrocyte lineage and molecular markers (genes or proteins) expressed under physiological and ASD conditions. Each human and mouse illustration indicates whether the molecular abnormality was found in clinical or experimental conditions [29,55,56,121,122,123,124].

**Table 1 brainsci-10-00951-t001:** Main changes observed in white matter regions of the ASD human brain. Age categorization was used to highlight the findings described in children and early adolescence (<15 years) vs. those observed in late adolescence and adults (>15 years). Asterisks indicate the brain regions where contradictory findings have been reported [21,30,36,60,64,65,67,70,71,72,73,74,75,76,77,78,79,80,81,82,83,84,85,86,87,88,89,90,91,92,93,94,95,96,97,98,99,100,101,102,103,104,105,106,107,108,109].

Age	Affected Brain Regions in ASD
<15 years	Increased white matter density: Supplementary motor area, left precentral, superior longitudinal fasciculus *, left cingulum *, right cingulate gyrus, prefrontal cortex *, radiate volume, corpus callosum *, right inferior frontal gyrus, putamen, insula, sublobar extranuclear area, right superior temporal gyrus, hippocampus, middle temporal gyrus *, fusiform gyrus, uncinate fasciculus *, inferior longitudinal fasciculus *, bilateral middle *, and left inferior cerebellar peduncle. Reduced white matter density: Superior longitudinal fasciculus *, cingulum *, cingulate gyrus, prefrontal cortex *, corona radiata, middle frontal gyrus, corpus callosum *, arcuate fasciculus, inferior frontal gyrus, forceps minor, fornix, anterior thalamic radiation, internal and external capsule, superior temporal gyrus, superior temporal sulcus, temporoparietal junctions, middle temporal gyrus *, right inferior frontal gyrus-middle temporal gyrus tracts, bilateral inferior frontal gyrus-fusiform gyrus tracts, inferior fronto-occipital fasciculus, occipitotemporal gyrus, uncinate fasciculus *, inferior longitudinal fasciculus *, amygdala, inferior temporal gyrus, bilateral superior, middle * and right inferior cerebellar peduncle, pontine crossing tracts and medial lemniscus, cerebellum, and corticospinal tract.
>15 years	Increased white matter density: Corpus callosum *, anterior and posterior thalamic radiation, right insula, bilateral amygdala-fusiform pathways temporal, left hippocampus-fusiform pathways, temporal segment of Superior longitudinal fasciculus *, right lingual gyrus, uncinate fasciculus, inferior fronto-occipital fasciculus *, inferior longitudinal fasciculus, and corticospinal tract *. Reduced white matter density: Superior longitudinal fasciculus, intraparietal sulcus, cingulum, anterior cingulate gyrus, right superior frontal gyrus, prefrontal cortex, middle frontal gyrus, corpus callosum *, left orbitofrontal cortex, inferior frontal gyrus, left putamen tracts, thalamus, forceps minor, anterior thalamic radiation, internal capsule, right hippocampus-fusiform pathway, inferior fronto-occipital fasciculus *, cerebellum, and corticospinal tract *.

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
