# Peer review of "Role of Oligodendrocytes and Myelin in the Pathophysiology of Autism Spectrum Disorder"

_brainsci, 2020, doi:10.3390/brainsci10120951_

Round 1

Reviewer 1 Report

Please see the attached document for suggestions on how to improve the manuscript.

Author Response

Dear Reviewer 1,

Sincerely, 

The authors

Reviewer 2 Report

The review is very well written and provides a lot of information. However there are some minor points.

1.The main points of this review is about the role of oligodendrocytes and myelin in autism. However the authors introduce Oligodendrocytes and myelin in section 3 extensively. There should be a short introduction about these in the introduction section as well.

2. A small cartoon depicting the role of oligodendrocytes and myelin  in normal brain function and how their malfunction leads to disease condition will be very helpful to the readers.

Author Response

Dear Reviewer 2,

Sincerely, 

The authors

Reviewer 3 Report

Galvez-Contreras et. al. reviewed the recent finding of the abnormal brain growth in white matter regions in autistic brains. The review described how brain growth varies with the age and how many of the alterations normalize throughout development. While some of the brain regions, like cerebellum, cingulum, and internal capsule remain hypo-myelinated. Which may link to the oxidative stress and differential expression of genes. From that, the authors link the oligodendrogenesis and neural myelination with the pathophysiology of ASD. This well-written and thorough review cover some major areas of ASD related brain disorganization.

While Recent findings showed the synaptic dysfunction is one of the major reasons for the pathophysiology of various psychiatric diseases. Abnormal functionality of glial cells leads to neuoinflamation and altering synaptic homeostasis.  

I found there is no mention of this major issue related to the neuo-glial interaction. I would recommend authors to include the gliotransmission, specifically how the volume regulated anion channels may cause synaptic dysfunction in ASD should be mentioned.

Also, there is no mention of the exaggerated expression of microglia and the probable role eIF4E in higher synapse density found in ASD.

I believe if authors decide to include those subjects in the review that will provide a more well-rounded review in this specific disorder.

Author Response

Dear Reviewer 3,

Sincerely, 

The authors
